

# Technical note: A comparative study of chemistry schemes for volcanic sulfur dioxide in Lagrangian transport simulations: a case study of the 2019 Raikoke eruption

Mingzhao Liu[1,2], Lars Hoffmann[1,2], Jens-Uwe Grooß[3,2], Zhongyin Cai[4], Sabine Grießbach[1,2], and Yi Heng[5]

[1]Jülich Supercomputing Centre, Forschungszentrum Jülich, Jülich, Germany
[2]Center for Advanced Simulation and Analytics (CASA), Forschungszentrum Jülich, Jülich, Germany
[3]Institute of Climate and Energy Systems, Stratosphere (ICE-4), Forschungszentrum Jülich, Jülich, Germany
[4]Yunnan Key Laboratory of International Rivers and Transboundary Eco-security, Institute of International Rivers and Eco-Security, Yunnan University, Kunming, China
[5]School of Computer Science and Engineering, Sun Yat-sen University, Guangzhou, China

**Correspondence:** Mingzhao Liu (mi.liu@fz-juelich.de)

**Abstract.** Lagrangian transport models are important tools to study the sources, spread, and life time of air pollutants. In order to simulate the transport of reactive atmospheric pollutants, the implementation of efficient chemistry and mixing schemes is necessary to properly represent the lifetime of chemical species. Based on a case study simulating long-range transport of volcanic sulfur dioxide ($SO_2$) for the 2019 Raikoke eruption, this study compares two chemistry schemes implemented in

the Lagrangian transport model Massive-Parallel Trajectory Calculations (MPTRAC). The explicit scheme represents first-order and pseudo-first-order loss processes of $SO_2$ based on prescribed reaction rates and climatological oxidant fields, i.e., the hydroxyl radical in the gas phase and hydrogen peroxide in the aqueous phase. Furthermore, an implicit scheme with a reduced chemistry mechanism for volcanic $SO_2$ decomposition has been implemented, targeting the upper troposphere and lower stratosphere (UT/LS) region. Considering non-linear effects of the volcanic $SO_2$ chemistry in the UT/LS region, we

found that the implicit solution yields a better representation of the volcanic $SO_2$ lifetime while the first-order explicit solution has better computational efficiency. By analysing the dependence between the oxidants and $SO_2$ concentrations, correction formulas are derived to adjust the oxidant fields used in the explicit solution, leading to a good trade-off between computational efficiency and accuracy. We consider this work to be an important step forward to support future research on emission source reconstruction involving non-linear chemical processes.

## 1 Introduction

Lagrangian transport models calculate trajectories of air parcels carrying trace gases and aerosols driven by external horizontal wind and vertical velocity fields. Lagrangian transport models are particularly useful for studying complex atmospheric processes, such as the interaction between different pollutants, chemical transformations, and the influence of different meteorological conditions. Lagrangian transport models have been widely applied in studies of natural and anthropogenic pollutant





emissions. Applications of Lagrangian models include the simulation of point source or regional pollutant emissions, such as
volcanic eruptions (Webster and Thomson, 2022), nuclear leakage accidents (Stohl et al., 2012), or wildfires (Evangeliou et al.,
2019). They can also be used for global chemical transport simulations and source estimations of greenhouse gases such as
carbon dioxide ($CO_2$) or methane ($CH_4$) (Bergamaschi et al., 2022; Che et al., 2022), long-lived gases of chlorofluorocarbons
(CFCs) or hydrofluorocarbons (HFCs) (Pommrich et al., 2014; Brunner et al., 2017), fossil emissions (Dalsøren et al., 2018),

and more. In contrast to grid-based Eulerian models, which represent fluid transport at fixed grid locations and suffer from
numerical diffusion due to limited grid resolution, trajectory-based Lagrangian models generally have much lower numerical
diffusion. In addition, Lagrangian models have the specific advantage of high scalability and computational efficiency, making
them suitable for long-range and large-scale simulations. When simulating a large number of chemical species, Lagrangian
models do not need to calculate transport separately for each species as in a grid-based Eulerian scheme (Brunner, 2012).

In the past, Lagrangian models have often been used to qualitatively identify the source and reproduce the spatial distribution
of plumes of trace gases, e. g., by tracking air parcels backward in time from receptors to sources to reconstruct the emissions
(Wu et al., 2017; Pardini et al., 2017). In case studies of volcanic eruptions, usually the effects of $SO_2$ lifetime variations
with altitude are neglected. However, Lagrangian models have the potential to be more quantitative. The essential step is to
accurately model the physical and chemical processes for accurate prediction of the species lifetime. For atmospheric species

such as $SO_x$, $NO_x$, hydrocarbons and halocarbons, the chemical reactions have a significant impact on estimating their sources
and sinks. Uncertainties introduced by chemical sinks can strongly influence the top-down source estimation (Stavrakou et al.,
2013).

In the Lagrangian framework, the motion of species is simplified to time integration along trajectories determined by meteo-
rological data. The modeling of species sinks is separated from the trajectory calculation and handled individually for each air

parcel. Following a rigorous Lagrangian approach, the time derivative of the mass or mixing ratio of each species is specified
with an explicit rate. This method can model processes such as dry/wet deposition, sedimentation, and first-order chemical
reactions. For example, in the study of Liu et al. (2023), the evolution of the $SO_2$ mass burden from the 2018 Ambae volcanic
eruption was simulated with explicit loss rates, including wet deposition, gas phase and aqueous phase reactions, and the height
sensitivity of the $SO_2$ loss rate, which is strongly related to cloud distributions, was discussed. To solve the second-order chem-

ical reactions, which depend on the concentration of the species, the loss rates are expressed by prescribing the OH and $H_2O_2$
concentrations using a monthly zonal mean climatology. However, this approach is more suitable for simulating long-lived
tracers at lower concentrations. In dense $SO_2$ plumes, the OH and $H_2O_2$ concentrations decrease rapidly and may even be
depleted by the reaction with $SO_2$. For more accurate simulations, these higher-order reactions should be considered.

To model non-linear chemical processes, an implicit chemical solver and an atmospheric mixing scheme should be consid-

ered (Brunner, 2012). In recent years, three main types of Lagrangian chemical and mixing schemes have been considered.
The first scheme uses a dynamically adaptive grid algorithm to mimic the stretching and distortion of air, such as in the
Chemical Lagrangian Model of the Stratosphere (CLaMS) (McKenna et al., 2002b, a) and the Alfred Wegener InsTitute LA-
grangian Chemistry/Transport System (ATLAS) (Wohltmann and Rex, 2009). The second scheme implements the chemistry
scheme on a fixed 3-D grid, assuming uniform mixing of individual species within the grid, such as in the U.K. Met Office's





Next-Generation Atmospheric Dispersion Model (NAME) (Redington et al., 2009) and the Hybrid Single-Particle Lagrangian Integrated Trajectory (HYSPLIT) (Stein et al., 2000) models. The third scheme implements the chemistry scheme on the Lagrangian air parcels, but uses a parameter to control the degree of mixing by adding a term to relax the trace gas concentrations in the air parcels to the background values averaged over a fixed 3-D grid, which is applied by the UK Meteorological Office (UKMO) chemistry transport model (STOCHEM) (Stevenson et al., 1998).

This paper focuses on the chemistry scheme implementation in the Lagrangian transport model Massive Parallel Trajectory Calculations (MPTRAC) (Hoffmann et al., 2016, 2022). Here, we selected the third approach outlined above, as it allows for online coupling of chemistry calculations with trajectory calculations, performed separately for each air parcel, followed by inter-parcel mixing, making it particularly suitable for large-scale parallelization. MPTRAC is designed for large-scale atmospheric simulations on HPC systems and has a hybrid Message Passing Interface (MPI) - Open Multi-Processing (OpenMP)

- Open Accelerator (OpenACC) parallelization scheme implemented, demonstrating excellent performance and scalability on both CPUs (Liu et al., 2020) and GPUs (Hoffmann et al., 2024b). The model has been used in several case studies of simulating long-range transport of volcanic $SO_2$, with emission estimates obtained from a backward trajectory method (Wu et al., 2017, 2018; Cai et al., 2022) and a more complex and compute intensive inverse modeling approach (Heng et al., 2016). In those studies, the height-dependent lifetime of $SO_2$ was not considered as an influencing factor, since older versions of

MPTRAC did not have a detailed chemistry scheme.

The main objective of this study is to introduce and assess a newly developed chemistry scheme with an implicit chemistry solver in the MPTRAC model, which improves long-range chemistry-transport simulations of volcanic $SO_2$ and allows for estimating the volcanic emission sources. We propose a small chemical mechanism with 12 species and 31 reactions to model the production and loss of OH, $HO_2$ and $H_2O_2$ in the UT/LS region, including reactions among $O(^1D)$, $O(^3P)$, H, OH, $HO_2$,

$O_3$ and $H_2O$, together with reactions of $SO_2$ with OH in the gas phase and oxidation with $H_2O_2$ in the aqueous phase. The aim is to model the dynamic OH and $H_2O_2$ fields as $SO_2$ oxidants to more realistically simulate and better represent the chemical lifetime of volcanic $SO_2$ in the UT/LS region. The chemical solver was built using the Kinetic Preprocessor (KPP) software package (Damian et al., 2002; Sandu and Sander, 2006). The KPP software provides a framework to automatically generate a Rosenbrock integrator for solving the stiff ordinary differential equations with specification of a chemical mechanism, including

the chemical equations, species, and rate coefficients. The KPP has been widely used in atmospheric chemical modeling, for instance in GEOS-CHEM (Henze et al., 2007) and MECCA (Sander et al., 2019).

In a case study, we conducted simulations of the June 2019 Raikoke volcanic eruption to compare and verify the modeled lifetime of the $SO_2$ emissions with TROPOspheric Monitoring Instrument (TROPOMI) satellite measurements. The Raikoke (48.17°N, 152.15°E) eruption was a notable event that has been discussed in the literature to some larger extent. The eruption

on 21-22 June was characterized by a series of explosive events that emitted $SO_2$ and volcanic ash into the lower stratosphere, impacting the stratospheric aerosol layer (Gorkavyi et al., 2021; Kloss et al., 2021). Cai et al. (2022) used MPTRAC to estimate the $SO_2$ emissions and to investigate the effects of the injection height and time and of the diffusion parameters. With an estimated amount of $(1.5 \pm 0.2)$ Tg of $SO_2$, the eruption was notable for being the largest $SO_2$ injection into the upper troposphere and lower stratosphere since the 2011 Nabro eruption (Cai et al., 2022). de Leeuw et al. (2021) discuss





the transport and chemical evolution of SO$_2$ emissions resulting from the 2019 Raikoke eruption, offering a comprehensive
comparison between NAME model simulations and TROPOMI observations.

The paper is organized as follows. In Sect. 2, we introduce the MPTRAC model, including the details of the implementation
of the chemistry schemes, and the TROPOMI SO$_2$ satellite data set. In Sect. 3, we discuss the results of the Lagrangian
chemical transport simulations with MPTRAC for the Raikoke eruption, including the evaluation of the modeled SO$_2$ fields in

the UT/LS region with the satellite data. Finally, Sect. 4 provides the summary and conclusions of the study.

## 2   Data and methods

### 2.1   Overview on the MPTRAC model

Massive-Parallel Trajectory Calculations (MPTRAC) is a Lagrangian transport model, which is designed for heterogeneous
CPU/GPU HPC systems and particularly suitable for large-scale and long-term atmospheric simulations in the free tropo-

sphere and stratosphere (Hoffmann et al., 2016, 2022). The model requires meteorological input fields, in particular the hori-
zontal wind and vertical velocity fields for the trajectory calculations, the temperature field for the chemistry calculations and
the cloud water fields for wet deposition and aqueous phase chemistry. The trajectories of the air parcels are calculated by using
the explicit mid-point method to maintain the balance between computational efficiency and accuracy (Rößler et al., 2018).
Following Stohl et al. (2005), diffusion is represented via stochastic perturbations added to the position of the air parcels and

subgrid-scale wind fluctuations are modelled as a Markov process using the Langevin equation. Unresolved convection is mod-
eled via the extreme convection parametrization (Gerbig et al., 2003; Hoffmann et al., 2023). A comprehensive description of
the model is given by Hoffmann et al. (2022). In the following section, we will discuss the two chemistry schemes implemented
in the model in more detail.

Lagrangian transport simulations with MPTRAC are driven by global meteorological reanalyses or forecasts. Here, we ap-

plied the European Centre for Medium-Range Weather Forecasts' (ECMWF's) fifth generation reanalysis ERA5 (Hersbach
et al., 2020), which provides hourly meteorological data at $0.3° \times 0.3°$ horizontal resolution on 137 vertical levels from the
surface up to 0.01 hPa. Utilizing a state-of-the-art data assimilation system, ERA5 integrates a vast array of historical observa-
tions, including satellite and in-situ data, to generate consistent, accurate, and temporally continuous records of meteorological
information from the year 1950 to the present. The ERA5 data provide significant improvements in meteorological informa-

tion compared to the previous generation ERA-Interim reanalysis, which benefits Lagrangian transport simulations (Hoffmann
et al., 2019).





## 2.2 Chemistry schemes and mixing in MPTRAC

### 2.2.1 First-order explicit chemistry scheme

In a Lagrangian model, linear decay processes over time $t$ can be expressed as

$$\frac{dy}{dt} = ky, \tag{1}$$

where $y$ represents the mass or volume mixing ratio (VMR) of a species in an air parcel. For a constant loss rate $k$, this ordinary differential equation is solved explicitly using the exponential equation,

$$y(t + \Delta t) = y(t)e^{-k\Delta t}. \tag{2}$$

Here, the rate coefficient $k$ is prescribed as a model input. In MPTRAC, various loss processes such as wet/dry deposition or sedimentation are modelled in this way. When solving first-order chemical reactions, e. g., photolysis, the explicit solution has good computational efficiency as it does not require numerical integration. The reactions of long-lived tracers with short-lived radicals can also be treated as pseudo-first-order reactions, assuming that production and loss of the radicals are in equilibrium and the pseudo-first-order rate constant $k$ can be expressed as the product of the second-order rate coefficient $k_{2nd}$ with the concentration of the oxidant, $k = k_{2nd}[X]$.

In MPTRAC, prescribed radical species concentrations are taken from a monthly zonal mean climatology of Pommrich et al. (2014), which has been prepared with the CLaMS model. Following Liu et al. (2023), the OH field is scaled with a correction factor based on the solar zenith angle to mimic diurnal variations while preserving daily averages. Similarly, the monthly zonal mean $H_2O_2$ background field was obtained from the Copernicus Atmosphere Monitoring Service (CAMS) reanalysis (Inness et al., 2019) compiled into a monthly zonal mean climatology. At each time step, background field values are interpolated from the tabulated zonal mean data according to the pressure and latitude of the air parcels.

Regarding the $SO_2$ chemical reactions, the gas phase oxidation with OH and the aqueous phase oxidation with $H_2O_2$ are considered. In the cloud aqueous phase, $H_2O_2$ oxidation is the predominant pathway when pH<5 (Seinfeld and Pandis, 2016; Rolph et al., 1992; Pattantyus et al., 2018). Other aqueous phase oxidation pathways such as ozone oxidation and catalyzed oxidation via Fe(III) and Mn(II) are neglected because of low pH values in highly concentrated $SO_2$ volcanic plumes. Liu et al. (2023) conducted a case study of the 2018 Ambae eruption with the first-order simplified chemistry scheme of MPTRAC, which provides more details.

### 2.2.2 Implicit chemistry scheme

A flow chart of the chemistry implementation in MPTRAC is shown in Fig. 1. The chemistry calculation follows the trajectory calculations and the mixing process. At each chemistry time step, the species VMRs of each air parcel are multiplied by the molecular density of air to convert them into concentration units (molecules/cm$^3$). After numerical integration of the chemical mechanism with the KPP integrator, the updated concentrations evolved in time are converted back to VMRs and assigned





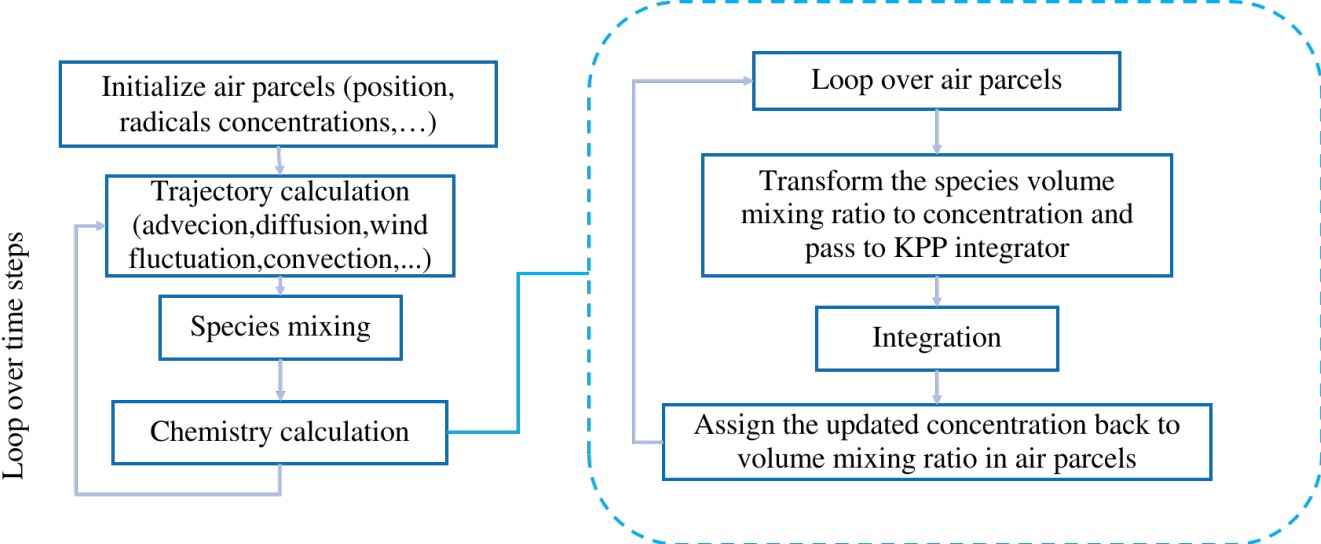

**Figure 1.** Simplified flow chart of the Massive-Parallel Trajectory Calculations (MPTRAC) Lagrangian chemistry-transport model.

back to each air parcel. The chemical processes of different air parcels are calculated independently of each other, making this an embarrassingly parallel compute problem, which is particularly suitable for parallelization.

Our proposed chemistry mechanism for volcanic $SO_2$ in the UT/LS region includes 31 reactions and 12 species, detailed

in Table 1. This mechanism models the dynamic production and loss of the OH and $H_2O_2$ concentration to further simulate $SO_2$ oxidation in both gas and aqueous phases, which are major loss mechanisms (Pattantyus et al., 2018; Rolph et al., 1992). OH production mainly occurs via the reaction of $H_2O$ with the excited oxygen radical $O(^1D)$ (Reaction R9). Its concentration exhibits a clear diurnal variation due to the strong relationship with solar radiation and photolysis, and its short lifetime from self-reaction and reactions with $O_3$ (Reaction R9) and $HO_2$ (Reaction R21) (Minschwaner et al., 2011; Tan et al., 2019). The

$H_2O_2$ is primarily formed through the self-reaction of hydroperoxy radicals ($HO_2$) (Reaction R24), which are mainly produced via H and $O_2$ (Rieger et al., 2018).

Note that this scheme for the oxidation of $SO_2$ was not designed for application in the boundary layer and lower troposphere, as it excludes nitrogen oxides (NOx), hydrocarbons, and carbon monoxide emissions and reactions. Furthermore, to simplify and constrain the chemistry calculations, each air parcel's ozone VMR is updated at every chemistry time step using ERA5

reanalysis data interpolation. Water vapor is also updated using ERA5 reanalysis data. The rate coefficients of all chemical reactions were taken from the current recommendations of Burkholder et al. (2019).

For photolysis modeling, we created 3-D photolysis rate look-up tables for the different species, $J = J(\theta_s, \text{TCO}, p)$, where $\theta_s$ is the solar zenith angle, TCO is total column ozone, and $p$ is pressure. The photolysis rate look-up tables cover 33 solar zenith angles from 0 to 96°, 8 total column ozone levels from 100 to 450 DU, and 66 pressure levels from 1013.25 to 0.1 hPa. The

look-up tables were generated using the DISSOC photolysis module of CLaMS, originally based on Lary and Pyle (1991) and improved over time in several studies (Becker et al., 2000; Hoppe et al., 2014; Pommrich et al., 2014). Note that for the DISSOC




**Table 1.** Proposed chemistry scheme for volcanic SO$_2$ oxidation in the UT/LS region.

| Index | Reaction |
|---|---|
| R1 | O($^3$P) + O$_2$ ⟶ O$_3$ |
| R2 | O($^3$P) + O$_3$ ⟶ 2 O$_2$ |
| R3 | O($^3$P) + OH ⟶ O$_2$ + H |
| R4 | O($^3$P) + HO$_2$ ⟶ OH + O$_2$ |
| R5 | O($^3$P) + H$_2$O$_2$ ⟶ OH + O$_2$ |
| R6 | O($^1$D) + O$_2$ ⟶ O($^3$P) + O$_2$ |
| R7 | O($^1$D) + O$_3$ ⟶ O($^3$P) + O$_2$ |
| R8 | O($^1$D) + H$_2$ ⟶ OH + H |
| R9 | O($^1$D) + H$_2$O ⟶ 2 OH |
| R10 | O($^1$D) + N$_2$ ⟶ O($^3$P) + N$_2$ |
| R11 | O($^1$D) + N$_2$ ⟶ N$_2$O |
| R12 | O($^1$D) + N$_2$O ⟶ prod |
| R13 | H + O$_2$ ⟶ HO$_2$ |
| R14 | H + O$_3$ ⟶ OH + O$_2$ |
| R15 | H + HO$_2$ ⟶ 2 OH |
| R16 | H + HO$_2$ ⟶ O($^3$P) + H$_2$O |

| Index | Reaction |
|---|---|
| R17 | H + HO$_2$ ⟶ H$_2$ + O$_2$ |
| R18 | OH + O$_3$ ⟶ HO$_2$ + O$_2$ |
| R19 | OH + OH ⟶ H$_2$O + O($^3$p) |
| R20 | OH + OH ⟶ H$_2$O$_2$ |
| R21 | OH + HO$_2$ ⟶ H$_2$O + O$_2$ |
| R22 | OH + H$_2$O$_2$ ⟶ H$_2$O + HO$_2$ |
| R23 | HO$_2$ + O$_3$ ⟶ OH + 2 O$_2$ |
| R24 | HO$_2$ + HO$_2$ ⟶ O$_2$ + H$_2$O$_2$ |
| R25 | O$_2$ + h$\nu$ ⟶ 2 O($^3$P) |
| R26 | O$_3$ + h$\nu$ ⟶ O($^1$D) |
| R27 | O$_3$ + h$\nu$ ⟶ O($^3$P) + O$_2$ |
| R28 | H$_2$O + h$\nu$ ⟶ H + OH |
| R29 | H$_2$O$_2$ + h$\nu$ ⟶ 2 O($^3$P) |
| R30 | SO$_2$ + H$_2$O$_2$(aq) ⟶ prod |
| R31 | SO$_2$ + OH ⟶ prod |

photochemistry model, we established the relationship between pressure, temperature, and ozone density via the U.S. Standard Atmosphere, 1976. The MPTRAC chemistry code determines $\theta_s$, TCO, and $p$ for each air parcel and linearly interpolates the corresponding $J$ values from the look-up tables. The look-up table approach has the advantages of high computational efficiency and easy implementation on both GPUs and CPUs.

### 2.2.3 Inter-parcel mixing algorithm

Atmospheric mixing is particularly important for properly modeling chemistry and transport in Lagrangian models (Brunner, 2012). The Lagrangian method inherently avoids numerical diffusion. A parametrisation of mixing needs to be included to simulate transport and chemistry in realistic manners. Here, we adapted the inter-parcel mixing scheme of Collins et al. (1997). This mixing scheme has the advantage of simple implementation and being well suited to CPU and GPU parallelization.

Following Collins et al. (1997), a relaxation term $d(c-\bar{c})$ is added to bring the VMR value $c$ of each air parcel closer to the average value $\bar{c}$ within fixed grid boxes. A mixing parameter of $d=0$ means there is no mixing whereas a mixing parameter of $d=1$ means the species VMR is fully relaxed to the grid box mean. The parameter $d$ controls the degree of mixing. Default values of $d$ are taken from Stevenson et al. (1998) to be $10^{-3}$ in troposphere and $10^{-6}$ in stratosphere. The grid box size was set to $5° \times 5° \times 1\,km$ (longitude $\times$ latitude $\times$ log-pressure height). Mixing was conducted at every time step of the model.



## 2.3 TROPOMI SO$_2$ observations

In order to validate the simulated volcanic SO$_2$ distributions and lifetime for the Raikoke eruption, we used the Tropospheric Monitoring Instrument (TROPOMI) SO$_2$ Level 2 data product (Veefkind et al., 2012; Theys et al., 2017). TROPOMI is an ultraviolet, visible, near and short-wavelength infrared spectrometer. It is mounted on ESA's Sentinel-5P satellite. The satellite operates in a near-polar Sun-synchronous orbit with a local time of 13:30 for the ascending nodes. TROPOMI samples the Earth's surface and atmosphere with a spatial resolution of $7 \times 3.5\,\mathrm{km}^2$ over a swath width of $2600\,\mathrm{km}$. TROPOMI monitors ozone, methane, formaldehyde, aerosol, carbon monoxide, NO$_2$ and SO$_2$ in the atmosphere.

The TROPOMI Level 2 data include three SO$_2$ data products that provide SO$_2$ total column densities derived assuming a $1\,\mathrm{km}$ deep SO$_2$ layer centered at 1, 7, and $15\,\mathrm{km}$ altitude. In this study, we used the $15\,\mathrm{km}$ product because it best fits the SO$_2$ mass release from Raikoke volcano into the UT/LS region. To obtain the SO$_2$ total mass burden, we first averaged the TROPOMI retrieval data over horizontal grid boxes of $0.1° \times 0.1°$ and then integrated the averaged values of the grid boxes. Note that the TROPOMI SO$_2$ total column data are provided in units of Dobson Units (DU). This is similar to ozone measurements, but for SO$_2$, $1\,\mathrm{DU}$ corresponds to a total column density of $2.85 \cdot 10^{-5}\,\mathrm{kg\,m}^2$. A lower bound of $0.35\,\mathrm{DU}$ was applied to both the satellite measurements and the model results to reduce the effect of noise from the satellite measurements and to improve comparisons.

## 3 Results of the Raikoke case study

### 3.1 Model initialization and setup

In an earlier study, Cai et al. (2022) investigate the time- and height-resolved SO$_2$ injection parameters of the 2019 Raikoke eruption based on a backward-trajectory approach and SO$_2$ retrievals from TROPOMI observations. In this work, we used the same estimates of the SO$_2$ injections as Cai et al. (2022) to initialize the release of air parcels in the transport simulations. A total SO$_2$ mass of $1.6\,\mathrm{Tg}$ was distributed over $10^6$ air parcels. The main injection peak occurs between June 21 and 22, 2019, at altitudes between 5 and $15\,\mathrm{km}$. The vertical profile of the SO$_2$ injection rates is shown in Fig. 2. Two numerical experiments were conducted with MPTRAC to compare the results of the linear explicit and non-linear implicit chemistry solutions. Here, the simplified explicit approximation scheme is similar to the work in Liu et al. (2023), considering the gas phase oxidation and aqueous phase oxidation with climatological background OH and H$_2$O$_2$ fields as introduced in Sect. 2.2.1. The KPP implicit solution considers the same reactions of SO$_2$, but the OH and H$_2$O$_2$ fields are modeled with the chemical mechanism listed in Table 1.

The simulated SO$_2$ plume evolution with the KPP implicit chemistry scheme is compared with TROPOMI observations in Figs. 3 and 4. From these comparisons, it is found that the MPTRAC model captures the overall distribution and movement of the volcanic SO$_2$ plume well during the first 10 days. Both the TROPOMI retrievals and the MPTRAC simulations show that the SO$_2$ plume continuously spreads over a larger area, with significant eastward motion and following the cyclonic flows. Similar to the studies of Cai et al. (2022) and (de Leeuw et al., 2021), the model results begin to lose the ability to capture





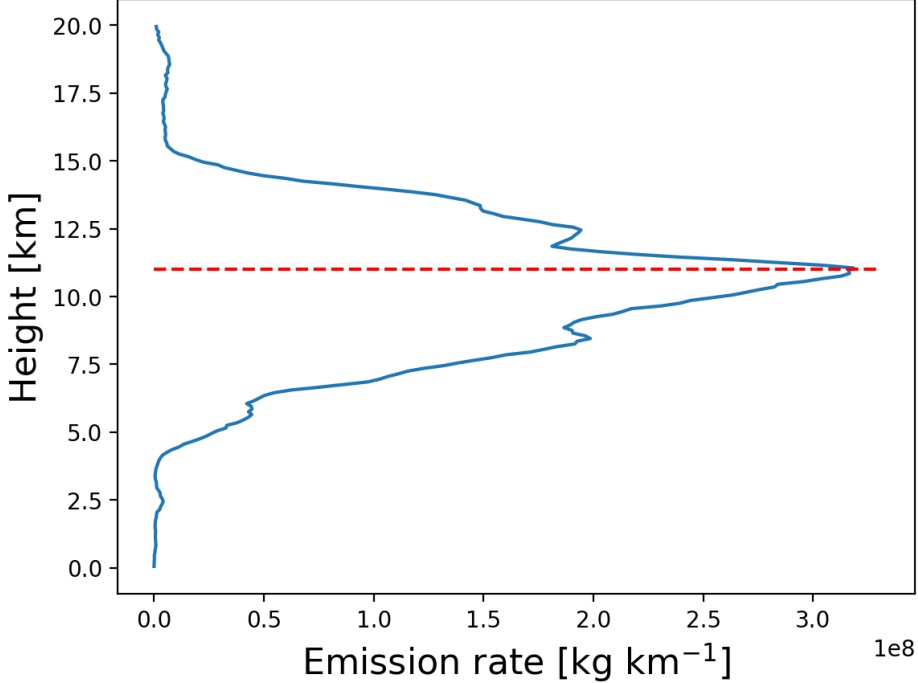

**Figure 2.** Vertical profile of SO$_2$ injections of the Raikoke eruption as estimated by Cai et al. (2022). The red line represents the tropopause height at the location of the volcano.

some structural features of the observed SO$_2$ distributions and show a stronger diffusion effect due to the limited resolution of the meteorological data after the first few days, but the overall propagation direction and the dispersion regions still remain consistent. In this paper, we will focus on the analysis of the volcanic SO$_2$ mass evolution and chemical lifetime based on this baseline simulation.

### 3.2 SO$_2$ chemical lifetime analysis

The time evolution of the total SO$_2$ mass burden of the 2019 Raikoke eruption from the MPTRAC simulations and TROPOMI observations is shown in Fig. 5. The total mass burdens from the model and the observations were obtained by integrating the SO$_2$ column densities on a $0.1° \times 0.1°$ horizontal grid. For comparison with the satellite data, the grid output of the model was sampled with a lower detection limit of 0.35 DU and weighted by the mean kernel function of the TROPOMI observations to reduce the impact of noise and other uncertainties and to account for the vertical sensitivity of the measurements. The SO$_2$ lifetime modeled with the implicit KPP solution is about 1.6 times longer than that of the simplified explicit approximation. This can be seen from the vertical profile of the e-folding lifetime in Fig. 6.

The lifetime profile obtained with the KPP implicit solution indicates that the SO$_2$ has much longer lifetimes at higher altitudes. The lifetime is typically within a few hours to several days below 5 km of altitude, while it exceeds 10 to 100 days





**Figure 3.** Evolution of SO$_2$ total column density distributions from 23 June 2019 to 30 June 2019 from the MPTRAC simulation with implicit chemistry scheme (right column) and TROPOMI satellite observations (left column). The red triangle marks the location of the Raikoke volcano.





**Figure 4.** Same as Fig. 3, but from 1 July 2019 to 7 July 2019.





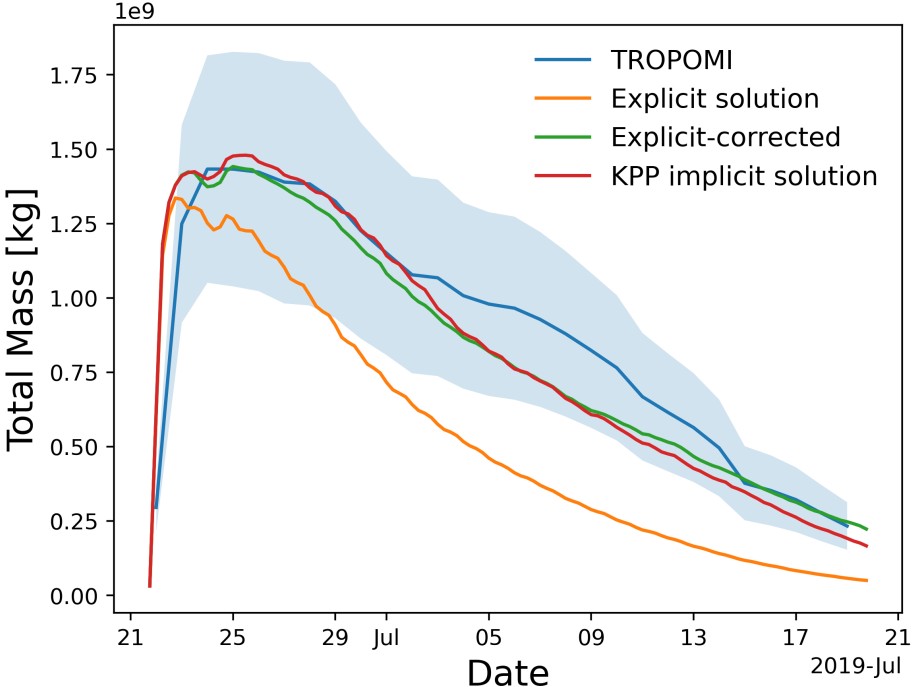

**Figure 5.** Temporal evolution of the SO$_2$ total mass burden of the 2019 Raikoke eruption from TROPOMI measurements (blue curve, shading shows uncertainty range) and MPTRAC simulations with explicit solution (orange curve), simplified explicit solution with correction (green curve), and KPP implicit solution (red curve).

above 5 km of altitude. Due to the frequent presence of clouds in the lower and middle troposphere, SO$_2$ is taken up into the liquid phase and decays rapidly due to cloud phase oxidation with H$_2$O$_2$ and wet deposition, leading to the shorter lifetime compared with the stratosphere. The vertical lifetime profile of the implicit KPP scheme shows a peak in the stratosphere that

is not found in the explicit scheme. This is attributed to a reduction of the OH concentration within the SO$_2$ plume, which is captured by the implicit chemistry scheme but not by the explicit scheme, which uses a constant OH climatology.

### 3.3   Correction for the simplified explicit scheme

From Figs. 5 and 6, it can be seen that the simplified explicit approximation overestimates the SO$_2$ decay rate by $\sim 60\%$ compared to the TROPOMI observations, whereas the implicit KPP solution agrees well with the TROPOMI observations.

Hence, there is also a $\sim 60\%$ difference between the explicit and KPP solution. This overestimation in the explicit scheme is likely due to changes in OH and other radicals resulting from the SO$_2$ injection itself, which are not accounted for in the climatology. To account for that, we perform a SO$_2$ dependent correction to the OH climatology. Essentially, the simplification of the explicit scheme applies a linear approximation to describe the non-linear oxidation processes. To compare the differences in lifetime of the two schemes, we performed another simulation without SO$_2$ injections to estimate the undisturbed background



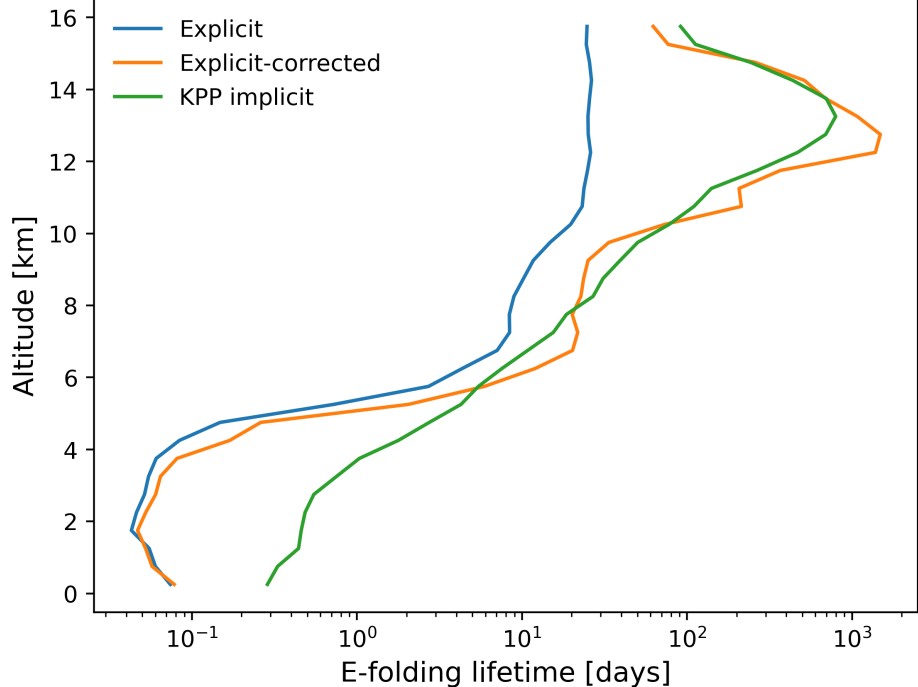

**Figure 6.** Vertical profiles of the average $SO_2$ e-folding lifetime of MPTRAC simulations with explicit solution (green curve), explicit solution with correction (orange curve) and KPP implicit solution (blue curve) during 25 June to 4 July.

levels of OH and $H_2O_2$ and compare them with the situation in the $SO_2$ plumes. Figure 7 shows the ratio of the OH and $H_2O_2$ concentrations in the $SO_2$ plume versus the background oxidant concentrations as a function of $SO_2$ VMR. The ratio decreases as the abundance of $SO_2$ increases, which means that the depletion of the oxidants is faster than their production. In highly-concentrated $SO_2$ regions the oxidants are almost completely depleted.

Note that the data points shown in Fig. 7 are filtered using a Z-score threshold, because there are factors other than the
chemical reactions that will affect the $SO_2$ distributions such as diffusion, convection, wet deposition, etc., causing the data not to be statistically reliable. After filtering outliers with low Z-scores, a correction formula to estimate OH concentrations in the plume from OH background levels and the $SO_2$ concentration can be obtained by regression,

$$[OH] = 4.72 \times 10^{-8} [SO_2]^{-0.83} \times [OH]_{background}. \tag{3}$$

Similarly, the correction formula for $H_2O_2$ is

$$[H_2O_2] = 3.13 \times 10^{-6} [SO_2]^{-0.57} \times [H_2O_2]_{background}. \tag{4}$$

By applying the corrections to the climatological OH and $H_2O_2$ data to account for their removal in the $SO_2$ plume, a simulation with the simplified explicit approximation with correction can be conducted. Figure 6 shows that above 10 km of altitude,





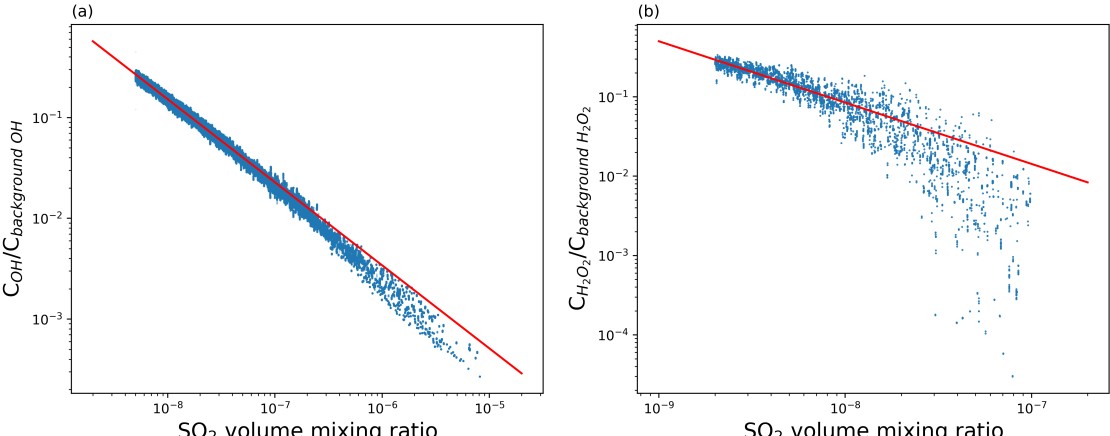

**Figure 7.** Ratio of OH concentration in dry atmosphere (a) and $H_2O_2$ concentration in cloud regions (b) in the Raikoke $SO_2$ plume versus background levels as a function of the $SO_2$ VMR. Each data point represents a single air parcel. The red curves represent the regression formulas given in Eqs. (3) and (4).

the correction for the simplified explicit solution shifts the simulated lifetime closer to the KPP implicit solution. The correction reduces the amount of the oxidants when $SO_2$ is highly abundant in the volcanic plume. The mass evolution simulated

by the corrected explicit solution is therefore in better agreement with the TROPOMI measurements and the KPP implicit solution (Fig. 5). These results reflect the fact that the chemical degradation of volcanic $SO_2$ is a non-linear process, which means that it should not simply be treated as pseudo-first-order reaction. The correction of the OH and $H_2O_2$ concentrations applied to the explicit solution improves the representation of $SO_2$ decay in the transport simulations, with almost no additional computational overhead.

## 260 3.4 Evaluation of the simulated OH field

The hydroxyl radical (OH) is an important oxidant in atmospheric chemistry, controlling the chemical decomposition of various species. For chemistry-transport simulations of volcanic $SO_2$ in the UT/LS region, oxidation via OH is the main factor of mass decay in the gas phase. In order to simulate the OH background conditions during the Raikoke case study without any enhanced levels of volcanic $SO_2$ abundance with the chemical mechanism introduced in Sect. 2.2.2, we distributed a total of 1 million air

parcels in between 9 to 17 km of altitude. The fourth generation Copernicus Atmosphere Monitoring Service (CAMS) global reanalysis data (Inness et al., 2019) were then used to compare with the OH field obtained with the implicit chemistry scheme implemented in MPTRAC. The CAMS reanalysis combines model data with observations using data assimilation techniques, providing a global distribution of atmospheric species.

Figure 8 shows a comparison of OH zonal mean distributions at different pressure levels from CAMS, MPTRAC, and the

CLaMS zonal mean climatology Pommrich et al. (2014). Both the CAMS reanalysis data and the CLaMS climatological data

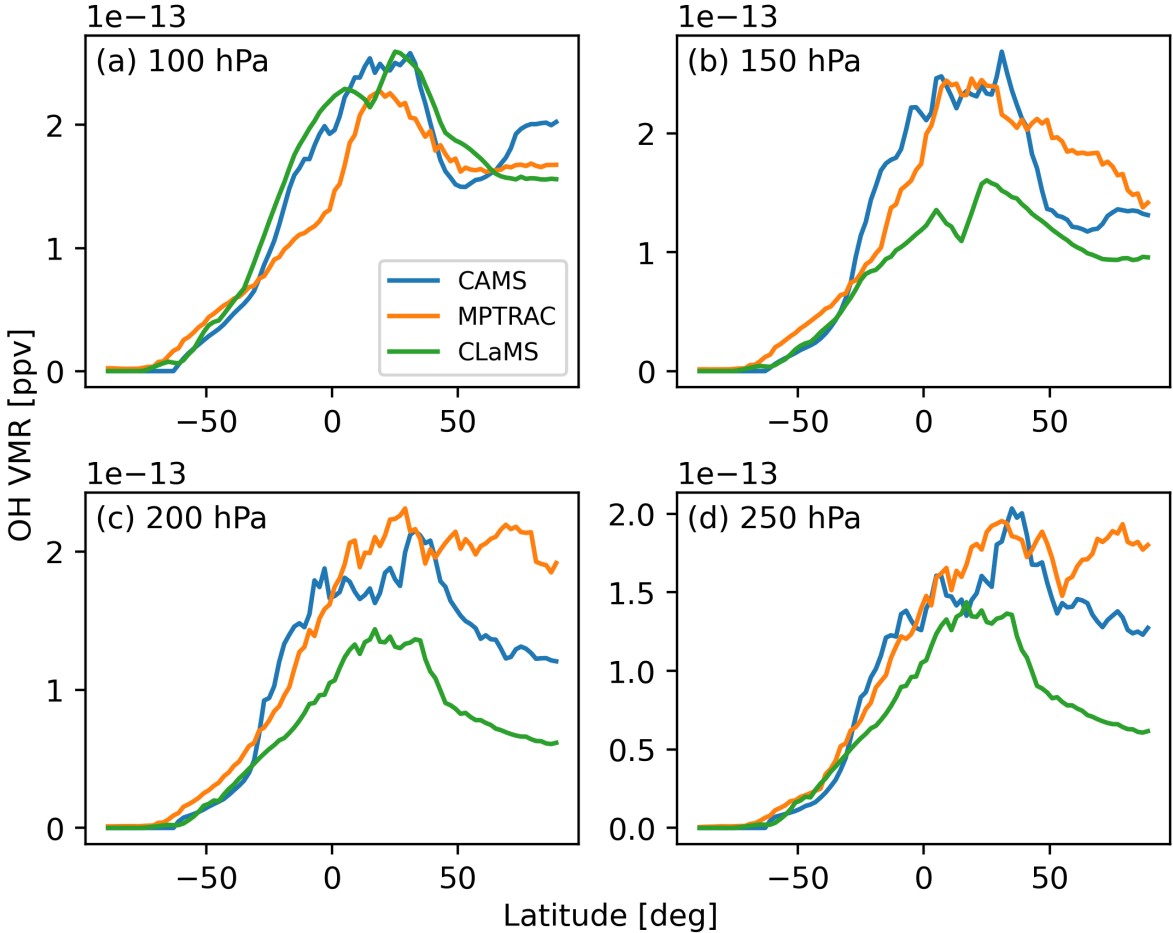

**Figure 8.** Zonal mean distributions of the OH field obtained from the CAMS reanalysis (blue curve), climatological data from the CLaMS model (green curve) and the MPTRAC simulation (orange curve) at pressure levels of 100 hPa (a), 150 hPa (b), 200 hPa (c) and 250 hPa (d) on 1 July 2019, 00:00 UTC.

have been evaluated by comparison with in-situ measurements in Liu et al. (2023). Furthermore, Fig. 9 shows the global CAMS and MPTRAC OH fields at pressure levels of $\sim 100\,\text{hPa}$, $\sim 150\,\text{hPa}$, $\sim 200\,\text{hPa}$ and $\sim 250\,\text{hPa}$ on 1 July 2019, 00:00 UTC. The OH fields obtained by MPTRAC show some similar distributions with respect to the solar zenith angle dependence as CAMS, which is due to the strong correlation between OH production and solar radiation. The OH fields simulated by the different models are at the same magnitude. The MPTRAC chemistry modeling can simulate the production and loss of the short-lived OH radical in the UT/LS region at reasonable accuracy, which is essential for $SO_2$ lifetime modeling.



**Figure 9.** Global OH fields obtained from the CAMS reanalysis (top) and MPTRAC simulations (bottom) at pressure levels of 100 hPa (a), 150 hPa (b), 200 hPa (c) and 250 hPa (d) on 1 July 2019, 00:00 UTC.



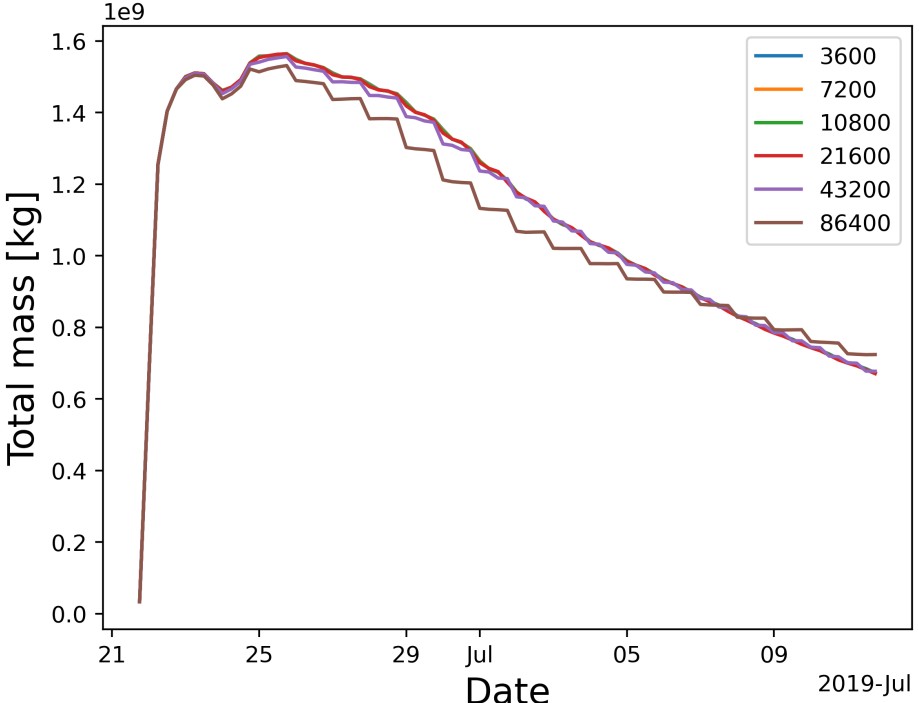

**Figure 10.** Total mass burden evolution simulated with different chemistry time step size settings of 3600, 7200, 10800, 21600, 43200 and 86400 seconds.

## 3.5 Selection of the chemistry time step

In MPTRAC, a fixed time step is applied for numerical integration of the trajectories and in most other modules. Based on previous studies (Rößler et al., 2018; Clemens et al., 2024), a time step of 180 s was selected for calculating trajectories with

the midpoint method and considering the spatiotemporal resolution of the ERA5 reanalysis data. This choice of the time step provides a reasonable trade-off in terms of accuracy and computational costs of the trajectory calculations, i.e., it is small enough to consider integration errors negligible while providing a minimum in computation time. In terms of computation effort, the time step of 180 s is acceptable for the explicit solution, but not efficient for the implicit chemistry calculations, which is particularly demanding in terms of computation.

To improve the computational efficiency, we implemented a distinct time step for the chemistry calculations. The chemistry time step is an important control parameter, as it has a significant impact on the computational efficiency. The chemistry time step size should be as large as possible without compromising the accuracy of the results. Figure 10 shows the $SO_2$ total mass time evolution obtained by different chemistry time steps. The results diverge when the time step of chemistry solution is larger than 6 h. Therefore, we recommend using a chemistry time step of 3 to 6 h to balance the computational efficiency and accuracy

of the solution for the volcanic $SO_2$ oxidation.




## 3.6 Assessment of computational costs

The KPP-based implicit chemistry solver implemented in MPTRAC can solve complex chemical mechanisms with flexible definition of the chemical reactions and species. In the case of the volcanic $SO_2$ chemistry-transport simulations, the corrected explicit solution achieves similar results with higher computational efficiency. For now, GPU offloading has been implemented using OpenACC for the simplified explicit scheme, while for the implicit scheme this is planned for the future. Figure 11 shows the runtime of MPTRAC simulations on the JUWELS supercomputer at the Jülich Supercomputing Centre comparing the implicit and explicit chemistry schemes using a chemistry time step of 21600 s and 180 s, respectively. The CPU simulations were conducted on a single compute node of the JUWELS Cluster (Jülich Supercomputing Centre, 2019), parallelized with 48 OpenMP threads. Each compute node of JUWELS Cluster contains 2 Intel Xeon Platinum 8168 CPUs with 24 physical cores per CPU. The GPU simulations were conducted on JUWELS Booster (Jülich Supercomputing Centre, 2021) with 2 AMD EPYC Rome 7402 CPUs and 4 NVIDIA A100 GPUs.

Regarding the chemistry, Figure 11a shows that the simplified explicit solution is about 5.3 times faster than the KPP implicit solution for a simulation with 1 million air parcels. Considering that file-I/O and meteorological data preprocessing also require a substantial part of the total runtime, the total runtime is reduced by $\sim 56\%$ by choosing the more efficient explicit chemistry scheme. The GPU solution of the simplified explicit chemistry scheme on JUWELS Booster reduces the total runtime by $\sim 75\%$, with the runtime required for chemistry becoming nearly negligible. As the scale of the simulation increases (i. e., for simulations with more air parcels), the advantage of the explicit solution in terms of computational efficiency becomes even more pronounced. Figure 11b shows that the total runtime is reduced by 92% for a simulation with 3 million air parcels. This improvement in computational efficiency is particularly relevant for large-scale chemistry transport simulations, where finding the proper trade-off between accuracy and computational costs is most critical. In the future, the KPP implicit scheme might also be equipped with GPU offloading to improve its performance (Alvanos and Christoudias, 2017; Christoudias et al., 2021).

## 4 Summary and conclusions

The Lagrangian transport model MPTRAC provides two alternative schemes to simulate $SO_2$ chemistry for volcanic eruptions. The explicit scheme uses monthly mean zonal climatology data of radical species to determine reaction rates and calculate mass loss as an exponential decay of $SO_2$. The implicit scheme uses the Rosenbrock integrator provided by the KPP software package to calculate the multi-species chemical reaction processes. KPP can solve complex chemical mechanisms with flexible definition of chemical species, reactions, and rate coefficients. The implicit solution allows for more complex, non-linear chemical mechanisms with dynamic reaction rates depending on the species concentrations. Here, we propose a chemical mechanism with 31 reactions and 12 species to model the production and loss of the short-lived OH radical and $H_2O_2$, which are essential for the decomposition of $SO_2$ in the gas and aqueous phase, respectively, in the UT/LS region. The mechanism proposed here can serve as a basis for developing more complex chemical mechanisms in future studies. It could also be further optimized by evaluating the relevance of each reaction pathway to reduce the computational effort.



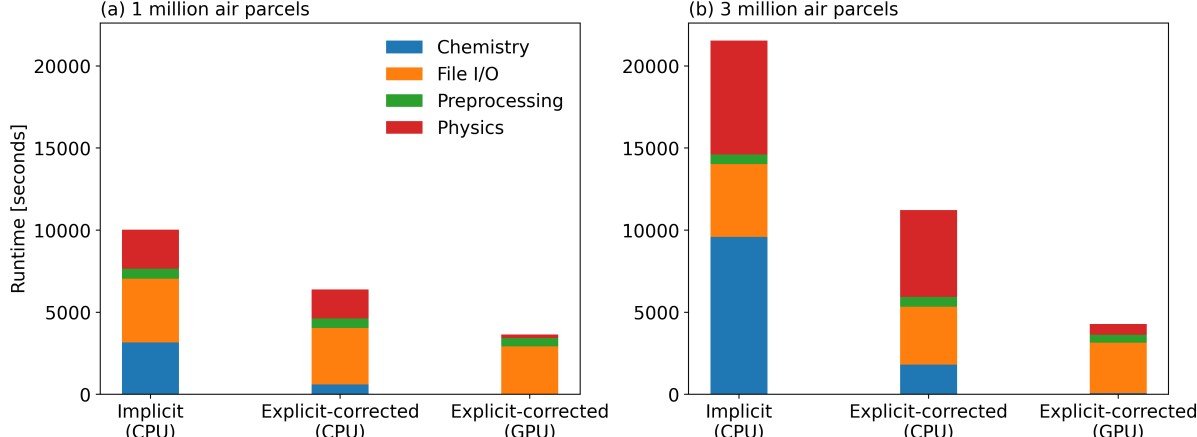

**Figure 11.** Computational costs of the Raikoke simulations with different chemistry schemes (the implicit scheme and the simplified explicit scheme with correction) on the JUWELS Cluster (CPU) and JUWELS Booster (GPU) for a simulation with 1 million air parcels on a scale of 1 million air parcels (a) and 3 million air parcels (b). The blue bar represents the runtime for the chemistry calculations, while the red bar represents the runtime for the other physical modules besides chemistry.

The OH radical is the main oxidant in the gas phase, which largely controls the decomposition of $SO_2$ in the UT/LS. We compared the OH radical field obtained by the proposed mechanism in MPTRAC with CAMS reanalysis data and climatolog-
ical data from the CLaMS model. The OH radical field obtained by the proposed mechanism shows a clear characteristic of diurnal variation and concentrations at a similar level compared to these reference data sets. The cloud phase $H_2O_2$ oxidation also has a significant impact on the $SO_2$ decay in the lower troposphere. It must be taken into account to properly represent the loss of tropospheric $SO_2$ over time, especially for other volcanic eruptions, such as those happening in the tropics with lower peak emission heights (Liu et al., 2023).

By applying the proposed chemical mechanisms in the implicit and explicit chemical solution to the case study of the June 2019 Raikoke eruption, the mass burden evolution and the vertical distribution of the lifetime are discussed. The implicit so-lution with the numerical integrator generated by KPP shows good agreement with the TROPOMI high-resolution satellite measurements with respect to the $SO_2$ total mass burden evolution. However, the explicit solution using first-order simplifi-cation shows an overestimation of the $SO_2$ decay rate by 60% compared to the implicit solution. This overestimation is due
to using OH and $H_2O_2$ climatological background field without considering enhanced levels of $SO_2$. By comparing the OH and $H_2O_2$ concentrations in the $SO_2$ plume with the background values, we obtained correction formulas $y = ax^b$ for OH and $H_2O_2$ by regression. The correction formulas are used to adjust the oxidant concentrations interpolated from the prescribed climatology data. This correction formula imposes a non-linear behavior on the chemical degradation of $SO_2$ and significantly improves the $SO_2$ lifetime representation in the UT/LS in the simulations with the explicit chemistry scheme. We tested the
correction also for another volcanic eruption case, i.e., the July 2018 Ambae eruption (Liu et al., 2023), where it yields a comparable total lifetime of $SO_2$ to that predicted by the KPP implicit solution.



The proposed simplified explicit solution with correction provides realistic chemical lifetime modeling compared to the implicit solution, but achieves more than 5 times higher computational efficiency in our example. For the trade-off between computational efficiency and simulation accuracy, the simplified explicit solution with correction is believed to have the capability to be applied in the future large-scale application requiring high computational effort, such as inverse modeling for emission reconstruction. The backward trajectory approach used in our previous studies of $SO_2$ volcanoes did not consider realistic chemical lifetime modeling. We are currently developing inverse modeling for source estimation, including support for nonlinear chemical mechanisms. The implications between lifetime modeling and source reconstruction are also being investigated. As a result of this work, the MPTRAC model has received significantly improved capabilities for Lagrangian chemistry modeling on current and upcoming HPC systems, opening up many opportunities for future studies and applications.

*Code and data availability.* The MPTRAC model (Hoffmann et al., 2016, 2022) is distributed under the terms and conditions of the GNU General Public License (GPL) version 3. The version 2.7 release of MPTRAC used in this paper is archived on Zenodo (Hoffmann et al., 2024a). Newer versions of MPTRAC are available from the repository at https://github.com/slcs-jsc/mptrac (last access: 15 July 2024). The ERA5 data were obtained from the European Centre for Medium-Range Weather Forecasts, see https://www.ecmwf.int/en/forecasts/datasets (Hersbach et al., 2020). The TROPOMI data (Theys et al., 2017) are provided via the Copernicus Data Space Ecosystem.

*Author contributions.* Conceptualization: M.L. and L.H.; Data curation: L.H., S.G., J.G. and Z.C.; Formal analysis: M.L.; Methodology: L.H., J.G. and Y.H.; Software: M.L. and L.H.; Supervision: L.H.; Validation: M.L.; Writing – original draft: M.L. and L.H.; Writing - review & editing: M.L., L.H., S.G., J.G., Z.C. and Y.H.

*Competing interests.* Jens-Uwe Grooß is a member of the editorial board of Atmospheric Chemistry and Physics.

*Acknowledgements.* This research was supported by the Helmholtz Association of German Research Centres (HGF) through the Joint Laboratory for Exascale Earth System Modeling (JL-ExaESM). The authors are grateful to the Jülich Supercomputing Center for providing computing time and storage resources on the JUWELS supercomputer. We acknowledge the use of the AI tool DeepL for language editing.



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
