# Peer review of "Technical note: A comparative study of chemistry schemes for volcanic sulfur dioxide in Lagrangian transport simulations: a case study of the 2019 Raikoke eruption"

_EGUsphere, 2024_

## Author Comment (AC1)

**Reply to editor comments**

Dear Referees,

Thank you for the time and effort spent on the manuscript. We considered all comments and hope that the revised draft properly addresses the open issues. Please find our point-by-point replies below (colored in blue). A revised manuscript with tracked changes has been uploaded.

Best regards,

Mingzhao Liu

**Referee #1**

The authors enhance the MPTRAC model by introducing and evaluating two chemistry schemes for simulating volcanic $SO_2$. The explicit scheme models first-order and pseudo-first-order $SO_2$ loss processes, while the implicit scheme, tailored for $SO_2$ transport in the UT/LS, employs a reduced chemical mechanism. The Raikoke eruption serves as a case study. The implicit scheme provides a more accurate representation of $SO_2$ lifetime, albeit at a higher computational cost. The paper is well-written, with clear figures supporting the text and conclusions. I believe the paper is publishable, pending the authors' response to my query regarding the implicit chemical mechanism. Specifically, I'm unclear about its derivation and validation. Are there any prior studies that have employed this mechanism?

We'd like to thank the referee regarding the positive evaluation of this study. We agree that a more detailed discussion of the derivation and validation of the proposed chemical mechanism for volcanic $SO_2$ decomposition in the UT/LS region will be helpful. We revised the manuscript accordingly. Please see reply to comment on line 149 below.

Minor questions/comments:

1st sentence. Not clear what is driven by wind and velocity fields, models or aerosols? Please rephrase.

We rephrased: 'Lagrangian transport models simulate the trajectories of air parcels that carry trace gases and aerosols through the atmosphere. These trajectories are determined using external datasets, which provide horizontal wind and vertical velocity fields obtained from meteorological reanalyses or forecast models.'

Line 83: reference on TROPOMI is missing

We added references to Veefkind et al. (2012) and Theys et al. (2017).

1st sentence in Sec. 2.2.2 does not correlate with what is shown in Fig. 1.

We rephrased: 'Figure 1 shows a simplified flow chart of the MPTRAC model, including details of the chemistry calculations.'

Line 149: please see my major question above.

We carefully revised Sect. 2.2.2, which introduces the implicit chemistry scheme and the proposed chemical mechanism for the oxidation of volcanic $SO_2$ in the UT/LS region. We added a statement that this particular mechanism is newly proposed and has not been published or evaluated elsewhere. We clarified that the mechanism has been constructed starting from OH oxidation of $SO_2$ in the gas phase and $H_2O_2$ oxidation in the aqueous phase and that it dynamically models the production and loss of OH and $H_2O_2$. We also noted that we included additional reactions, even though they are less relevant to the primary focus, to capture potential secondary effects that may affect the system under certain conditions or to improve applicability.

Table 1: what is prod?

We replaced 'prod' by 'products' to avoid shorthand terminology, which represents products that are not explicitly included in this chemical scheme.

Line 184: please expand 'ESA'?

Expanded to 'European Space Agency'.

Line 208: Please include evolution of $SO_2$ plume evolution computed using explicit and corrected explicit schemes in Fig 3 and 4.

For comparison with Figs. 3 and 4 in the paper, Figs. 1 and 2 in this reply illustrate the $SO_2$ total column density distributions from June 23 to July 7, 2019, based on MPTRAC simulations using the uncorrected and corrected explicit chemistry schemes. Overall, the shapes of the volcanic $SO_2$ plumes across the three MPTRAC simulations are quite similar, as expected, since the advection and diffusion calculations were consistent across all simulations. However, the magnitudes differ slightly, with the explicit chemistry scheme producing lower $SO_2$ values compared to the explicit-corrected and implicit chemistry schemes. This difference is anticipated, as the explicit scheme tends to overestimate $SO_2$ loss from OH oxidation by relying on fixed OH concentrations from climatology. We refrained from adding these additional figures in the paper, but included the following statement: 'The explicit scheme closely matches the implicit scheme in plume shape (not shown) but shows overall lower $SO_2$ column densities due to higher OH oxidation loss rates.'

Line 209: maybe replace 'movement' by 'transport'?

We rephrased: 'Based on these comparisons, the MPTRAC model effectively captures the dispersion and transport of the volcanic $SO_2$ plume during the first 10 days.'

Line 212: '...the model results begin to lose...'. Please use different wording.

We rephrased: 'Consistent with the findings of Cai et al. (2022) and de Leeuw et al. (2021), the model results gradually lose the ability to capture certain structural features

[Figure]

Figure 1: Evolution of SO₂ total column density distributions from 23 June 2019 to 29 June 2019 from the MPTRAC simulation with the explicit-corrected chemistry scheme (left column) and explicit (uncorrected) chemistry scheme (right column). The red triangle marks the location of the Raikoke volcano.

[Figure]

Figure 2: Same as Fig. 1, but from 1 July 2019 to 7 July 2019.

of the observed $SO_2$ distributions after the first few days. This decline is attributed to the limited resolution of the meteorological data, which also leads to a stronger diffusion effect. Nevertheless, the overall propagation direction and dispersion regions remain well-aligned with observations.'

Line 240: Please rephrase the 1st sentence (situation in the $SO_2$ plumes?)

We rephrased to 'conditions in the $SO_2$ plume.'

Line 258: 'transport' simulations?

We replaced this by 'chemistry-transport simulations'.

Line 267: Please specify which gases CAMS assimilates. I do not think that is assimilates OH and $H_2O_2$.

We added: 'Key species assimilated in the CAMS reanalysis include ozone ($O_3$), carbon monoxide (CO), nitrogen dioxide ($NO_2$), methane ($CH_4$), and carbon dioxide ($CO_2$). Note that the hydroxyl radical (OH) and hydrogen peroxide ($H_2O_2$) are not directly assimilated from observations; instead, their concentrations are computed using the chemistry scheme implemented in the CAMS reanalysis.'

Line 271: Please remove Fig. 8 or add analysis of what is shown there.

We expanded the description of the figure: 'Figure 8 compares zonal mean OH distributions at different pressure levels from the CAMS reanalysis, the MPTRAC model, and the CLaMS climatology. The best agreement among the three datasets is observed at the 100 hPa pressure level. Zonal mean OH concentrations increase gradually from near zero under polar winter conditions at high southern latitudes to approximately $2.5 \times 10^{-13}$ ppv in the northern subtropics. At northern mid and high latitudes, OH concentrations remain within 1.5 to $2 \times 10^{-13}$ ppv. At lower vertical levels (150 to 250 hPa), differences of up to a factor of two are observed in the tropics and northern mid- to high latitudes. Overall, the CAMS and MPTRAC datasets tend to agree more closely with each other than with the CLaMS data. Differences in OH concentrations arise from variations in chemical mechanisms, emissions, meteorological conditions, photolysis schemes, spatial resolution, boundary conditions, and the representation of stratosphere-troposphere exchange processes.'

Fig 9. Not clear on MPTRAC figures. How were they obtained? Is it a $SO_2$ free run? Is it implicit, explicit or explicit with correction chemistry?

We revised the description of the figure and added: 'The OH fields obtained by MPTRAC were generated using its implicit chemistry scheme under $SO_2$-free conditions.'

Line 340: Please include this correction test in the manuscript. It will strengthen the publication.

The application of the correction for the Ambae case has already been discussed by Liu et al. (2023). Figure 3 in this reply shows the $SO_2$ total mass curves for the Ambae

[Figure]

Figure 3: SO$_2$ total mass curves from MPTRAC simulations and TROPOMI observations for the July 2018 Ambae eruption. See Liu et al. (2023) for reference.

eruption for reference. We revised and expanded on the comparison with the Ambae case as follows: 'We also applied the correction to another volcanic eruption, the July 2018 Ambae eruption (Liu et al., 2023), and observed that it produced a total SO$_2$ lifetime comparable to that predicted by the KPP implicit solution. Notably, despite differences in geographic and atmospheric condition – Ambae being a tropical eruption and Raikoke occurring in Northern Hemisphere mid- and high latitudes – the same parameter values for $a$ and $b$ in the correction formula were used. This suggests that the correction approach is transferable across diverse eruption scenarios.'

Sect. 4: Summary and conclusions can be shortened and avoid repetitions.

We revised the summary and conclusions as suggested.

**Referee #2**

Overview

The paper presents a study of using three approaches to simulate chemical loss of the volcanic SO$_2$ with the Lagrangian MPTRAC model for the 2019 Raikoke eruption. The authors use a) an interactive chemistry scheme of 31 gas phase an photolysis reaction,

b) prescribed OH and $H_2O_2$ monthly mean fields calculated by the CLAMS model and c) a variant of b scaled to match the results of a). Retrievals of the volcanic $SO_2$ from TROPOMI are used as reference to estimate the decay of the emitted $SO_2$, and to provide the source term. The OH fields are quantitatively compared against the OH fields from the CAMS reanalysis. Scheme a) performs best and c) can mostly replicate a) at a much-reduced computational cost.

General remarks

The paper is well written and informative. But, It remains a technical note in nature as it mainly compares the computational cost and relatively briefly the scientific results of the comparison of the chemistry schemes, without going into much detail about specific aspects and issues of the simulation. For example a longer discussion comparing the impact of the reaction with $H_2O_2$ associated to wet removal and the loss by OH gas phase chemistry could have been interesting. Other aspects of the volcanic $SO_2$ modelling such as mixing and transport could have been of interest too. Finally, the resulting stratospheric sulphate is not discussed at all.

We sincerely thank the referee for the thoughtful and constructive comments on our paper. We are pleased to hear that you found the study well-written and informative. We appreciate the observation that the paper primarily focuses on the technical comparison of the chemistry schemes, and we agree that a deeper exploration of specific aspects such as the roles of wet removal and sulfate formation would indeed enrich the discussion. In response to your suggestions, we have incorporated additional details in the manuscript as outlined below. Given the scope of this study, our primary aim was to emphasize the computational trade-offs and accuracy of the different chemistry schemes. However, we are committed to addressing additional aspects in follow-up studies.

Specific remarks

L24 Please add also some disadvantages of Lagrangian dispersion model here.

We added: 'However, Lagrangian models can face challenges such as complex boundary condition handling, difficulties in representing concentration fields and physical diffusion, high particle requirements for dense regions and uneven computational loads.'

L44 Please clarify second-order, do you mean bimolecular reactions

The term second-order reactions in our context does not necessarily mean bimolecular reactions specifically, though it often refers to them. Here, second-order reactions describe processes where the reaction rate depends on the product of the concentrations of two reactants. For example, the gas-phase reaction of $SO_2$ with OH to form $HSO_3$ is bimolecular and therefore second-order, as its rate is proportional to $[SO_2] \cdot [OH]$. In the paragraph, we aimed to highlight that, under certain conditions – such as within dense volcanic plumes – the simplifying assumption of prescribing static oxidant fields (like monthly climatologies for OH and $H_2O_2$) may not adequately capture the dynamic behavior of second-order reactions. This limitation arises because the oxidant concentrations can decrease or deplete

due to reactions with $SO_2$. For accurate modeling in such cases, dynamic coupling between the reactant concentrations and reaction rates is necessary to properly represent these higher-order interactions. We revised the text to ensure it conveys this distinction more precisely.

L49 Please clarify why a mixing scheme is required for the chemistry.

We added: 'In essence, while Lagrangian models efficiently track particle trajectories, a mixing scheme is essential to accurately represent diffusion and produce smooth gradients in concentration fields. Without a mixing scheme, particles carrying different reactants may remain isolated, resulting in pronounced local irregularities and inaccurate reaction rates. Mixing schemes address these issues by smoothing out irregularities in particle distributions, improving physical realism and ensuring numerical stability.'

L53 You talk about mixing only for the first group. Is it not important for the other types?

All three schemes described here account for mixing. In the second and third groups of schemes, mixing is implemented by averaging particle data over a regular grid and adjusting individual air parcel concentrations to align with the grid-averaged values.

L130 Please clarify the differences between MPTRAC and CLAMS (here or in the introduction)

We added: 'In contrast to MPTRAC, which focuses on efficient particle transport with limited or simplified chemistry, the CLaMS model emphasizes detailed chemical reaction schemes for stratospheric processes. We consider the CLaMS model a reliable source for creating radical species climatologies for the UT/LS region.'

L148 'embarrassingly' – I am not sure if this is the right word in this context.

We rephrased the sentence: 'The chemical processes of different air parcels are calculated independently of each other, making this an ideal parallel compute problem, which is particularly suitable for parallelization.'

L193 Perhaps that detail on the unit conversion does not need to be mentioned here.

We suggest keeping this information here, as not all readers may be familiar with the specifics of the definition of Dobson units for $SO_2$, particularly with respect to the differences compared to the definition for ozone, for example.

L219 Please comment on the change in the $SO_2$ decay rate derived from S5P from about 3.7.-15.7. (Fig 6)

We added: 'While still within the uncertainty range of the TROPOMI observations, Fig. 6 highlights a notable discrepancy between the $SO_2$ total mass from TROPOMI and the MPTRAC simulations during 3–15 July 2019. de Leeuw et al. (2021) attributed the change in $SO_2$ lifetime observed by TROPOMI about 10 days after the eruption to the removal of a large fraction of the tropospheric $SO_2$ mass through aqueous phase oxidation and wet deposition. Beyond this point, the TROPOMI observations becomes dominated by

the stratospheric component of the $SO_2$ cloud. Since the stratosphere contains much less moisture, $SO_2$ removal occurs at a much slower rate, primarily through gas phase reactions with OH, leading to significantly longer e-folding lifetimes. Our MPTRAC simulations are based on the $SO_2$ injection profiles provided by Cai et al. (2022), which may underestimate the stratospheric fraction and, consequently, the impact on long-range transport. However, the estimates of $SO_2$ mass loss from the MPTRAC simulations (Fig. 5b) indicate a significant reduction in loss rates from wet deposition and $H_2O_2$ oxidation after July 3. This aligns closely with the findings of de Leeuw et al. (2021).'

L229 Please explain in more detail here or in the model description if and how cloud uptake was considered. It is only mentioned in the description for the explicit scheme. Is wet removal modelled in all scheme with the same method?

We agree that this aspect requires clarification and further explanation. To address this, we added a new Section 2.2.3 in the model description, which outlines the treatment of wet deposition in the MPTRAC model. Additionally, we clarified that this loss mechanism was considered in all the simulations covered by this study.

L259 Please discuss whether this correction could be used also for other cases, or if it is specific to the Raikoke case.

Following also a comment given by Referee #1, we revised and expanded on the Ambae case as follows: 'We also applied the correction to another volcanic eruption, the July 2018 Ambae eruption (Liu et al., 2023), and observed that it produced a total $SO_2$ lifetime comparable to that predicted by the KPP implicit solution. Notably, despite differences in geographic and atmospheric condition – Ambae being a tropical eruption and Raikoke occurring in Northern Hemisphere mid- and high latitudes – the same parameter values for $a$ and $b$ in the correction formula were used. This suggests that the correction approach is transferable across diverse eruption scenarios.'

L275 Please comment on the lack of the CAMS "OH plumes" over Japan and the Pacific in the MPTRAC fields.

We added the following discussion on the OH plumes over Japan and the Pacific: 'A closer inspection reveals distinct features in the CAMS OH fields that are absent in the MPTRAC fields, such as localized OH plumes over Japan and the Pacific visible in the CAMS data. This discrepancy likely stems from the simplified oxidant mechanisms used in MPTRAC, which smooth out regional OH enhancements captured by CAMS. Additional factors, such as differences in spatial and temporal resolution and the representation of regional emissions and environmental conditions, further contribute to the observed differences. Despite these limitations, the MPTRAC scheme demonstrates a reasonable ability to simulate the production and loss of the short-lived OH radical in the UT/LS region, offering a practical balance between accuracy and computational efficiency for modeling $SO_2$ lifetimes in volcanic plumes and other atmospheric scenarios.'

L329 Please clarify if you took this into account and try to quantify the impact of this

process on the SO$_2$ removal and lifetime.

To clarify, we added: 'Cloud phase oxidation has been considered in all cases in this study.' In Fig. 5, we added a new panel showing estimates of SO$_2$ total mass loss of the Raikoke eruption due to OH and H$_2$O$_2$ oxidation as well as wet deposition. These estimates were obtained from the simplified explicit solution with correction. The discussion of the figure was updated accordingly.

**References**

Cai, Z., Griessbach, S., and Hoffmann, L.: Improved estimation of volcanic SO$_2$ injections from satellite retrievals and Lagrangian transport simulations: the 2019 Raikoke eruption, Atmos. Chem. Phys., 22, 6787–6809, doi: 10.5194/acp-22-6787-2022, 2022.

de Leeuw, J., Schmidt, A., Witham, C. S., Theys, N., Taylor, I. A., Grainger, R. G., Pope, R. J., Haywood, J., Osborne, M., and Kristiansen, N. I.: The 2019 Raikoke volcanic eruption – Part 1: Dispersion model simulations and satellite retrievals of volcanic sulfur dioxide, Atmos. Chem. Phys., 21, 10 851–10 879, doi: 10.5194/acp-21-10851-2021, 2021.

Liu, M., Hoffmann, L., Griessbach, S., Cai, Z., Heng, Y., and Wu, X.: Improved representation of volcanic sulfur dioxide depletion in Lagrangian transport simulations: a case study with MPTRAC v2.4, Geosci. Model Dev., 16, 5197–5217, doi: 10.5194/gmd-16-5197-2023, 2023.

Theys, N., De Smedt, I., Yu, H., Danckaert, T., van Gent, J., Hörmann, C., Wagner, T., Hedelt, P., Bauer, H., Romahn, F., Pedergnana, M., Loyola, D., and Van Roozendael, M.: Sulfur dioxide retrievals from TROPOMI onboard Sentinel-5 Precursor: algorithm theoretical basis, Atmos. Meas. Tech., 10, 119–153, doi: 10.5194/amt-10-119-2017, 2017.

Veefkind, J., Aben, I., McMullan, K., Förster, H., de Vries, J., Otter, G., Claas, J., Eskes, H., de Haan, J., Kleipool, Q., van Weele, M., Hasekamp, O., Hoogeveen, R., Landgraf, J., Snel, R., Tol, P., Ingmann, P., Voors, R., Kruizinga, B., Vink, R., Visser, H., and Levelt, P.: TROPOMI on the ESA Sentinel-5 Precursor: A GMES mission for global observations of the atmospheric composition for climate, air quality and ozone layer applications, Remote Sens. Environ., 120, 70–83, doi: 10.1016/j.rse.2011.09.027, the Sentinel Missions - New Opportunities for Science, 2012.

---

## Author Response (AR2)

**Reply to editor comments**

Dear editor,

Thank you for your positive feedback and for recommending our manuscript for publication subject to minor revisions. We appreciate the constructive comments from the reviewers and have carefully addressed their concerns. Below, we outline the specific changes made in response to the remaining points.

Best regards,

Mingzhao Liu

**Response**

Concerning Referee #1's comment on line 212 of the original manuscript, I interpret the referee's meaning to be that this statement should be rephrased to remove the text "lose the ability". The model does not change, it is a static thing, so it cannot gain or lose abilities while running. If a model is inaccurate, then such inaccuracies may become more apparent as a simulations progresses, or under certain situations (e.g., certain areas in parameter space). Your statement should be made more precise here.

We agree with the reviewer's observation that the phrase "lose the ability" is not appropriate for describing a static model. To address this, we have rephrased the statement to more accurately reflect the behavior of the model. The revised text now reads:

"The accuracy of the model results in reproducing the structural features of the observed $SO_2$ distributions gradually decreases after 15 days of simulation time due to error propagation."

Concerning Refree #2's comment regarding line 259 of the original manuscript, in your edit, please be clearer about the location of Raikoke – it cannot be in both the mid- AND high-latitudes. Also, please be careful with grammar regarding volcanoes vs. eruptions, e.g., here, Ambae is a tropical volcano, not eruption. I might suggest test here similar to: "Ambae is situated in the tropics while Raikoke is in the mid-latitudes..."

We thank the reviewer for pointing out the need for greater precision regarding the location of Raikoke and the distinction between volcanoes and eruptions. We have revised the text as follows:

"Ambae is situated in the tropics and Raikoke is located in the mid-latitudes."